# Impact of Magnetohydrodynamics on Thermal Mixing Efficiency and Entropy Generation Analysis Passing Through a Micromixer Using Non-Newtonian Nanofluid

**DOI:** 10.3390/mi17010066

**Published:** 2025-12-31

**Authors:** Naas Toufik Tayeb, Youcef Abdellah Ayoub Laouid, Ayache Lakhdar, Telha Mostefa, Sun Min Kim, Shakhawat Hossain

**Affiliations:** 1Gas Turbine Joint Research Team, Ziane Achour University, Djelfa 17000, Algeria; t.naas@univ-djelfa.dz; 2Department of Mechanical Engineering, Ziane Achour University, Djelfa 17000, Algeria; 3Laboratory of Electro-Mechanical Systems, The Engineers National School of Sfax, University of Sfax, Sfax 3038, Tunisia; 4Department of Mechanical Engineering, Inha University, 100, Inha-ro, Michuhol-gu, Incheon 22212, Republic of Korea; 5Department of Biological Sciences and Bioengineering, Inha University, 100, Inha-ro, Michuhol-gu, Incheon 22212, Republic of Korea; 6Biohybrid Systems Research Center, Inha University, 100, Inha-ro, Michuhol-gu, Incheon 22212, Republic of Korea; 7Department of Industrial and Production Engineering, Jashore University of Science and Technology, Jessore 7408, Bangladesh

**Keywords:** chaotic advection, micromixer, mixing index, energy cost of mixing, entropy generation, magnetohydrodynamics

## Abstract

The present paper investigates the steady laminar flow and thermal mixing performance of non-Newtonian Al_2_O_3_ nanofluids within a two-layer cross-channel micromixer, employing three-dimensional numerical simulations to solve the governing equations across a low Reynolds number range (0.1 to 50). It also addresses secondary flows and thermal mixing performance with two distinct inlet temperatures for thin nanofluids. Additionally, it explores how fluid properties and varying concentrations of Al_2_O_3_ nanoparticles impact thermal mixing efficiency and entropy generation. Simulations were conducted to optimize performance by adjusting the power law index (*n*) across different nanoparticle concentrations (1–5%). The findings show that magnetohydrodynamics can enhance mixing efficiency by generating vortices and altering flow behavior, providing important guidance for improving microfluidic system designs in practical applications.

## 1. Introduction

The miniaturization of chemical and biological analysis devices has generated considerable interest in microfluidics. Numerous benefits are associated with the use of microfluidic systems, such as decreased reagent usage, accelerated response times and better control over reaction conditions, and portability [1]. Among the most critical aspects of microfluidic operations is efficient mixing at the microscale, where laminar flow regimes predominate due to low Reynolds numbers [2]. This limitation requires the implementation of highly effective mixing strategies to ensure complete and rapid homogenization of fluids within the limited dimensions of a microchannel.

Microfluidics [3] is an active field of research in the United States, Europe, Japan, and China. The global scientific community is showing sustained interest in the fabrication of microfluidic chips, in order to explore their analytical potential, adapt research methods to a reduced scale and synthesize new nanoscale systems within the chips. Polymers and surfactants are essential components of living materials and chemical industry processes, such as gases, nuclear proteins, detergents, oil recovery enhancement zones and drug delivery systems. They can easily form nanoparticles by self-organizing in microfluidic devices.

Microfluidic devices can be classified into two categories: passive and active devices [2]. In contrast to passive micromixers, which operate without external energy by utilizing channel geometry and pressure-driven flow to induce chaotic advection and enhance molecular diffusion, resulting in simple, robust, and easily integrated systems, active micromixers rely on external energy sources such as thermal [3], pressure [4], electric [5], magnetohydrodynamic [6], or acoustic fields [7] to actively perturb the fluids and accelerate mixing. This capability allows for greater dynamic control and mixing efficiency but introduces disadvantages, including more complex fabrication and reliance on external power during operation.

It should be noted that many review articles have presented different types of passive micromixers as well as their mixing mechanisms [8]. These devices are distinguished by variations in their dimensions and operating conditions, including the Reynolds number. Nevertheless, the literature remains largely descriptive, and there is a lack of quantitative comparisons of their mixing performances. The latter could prove valuable to designers, who are looking for information on the efficiency of different micromixers under consistent geometric and operational conditions [9].

Thus far, quantitative comparisons have concentrated on certain categories of passive micromixers. Falk and Commenge [10] performed an extensive investigation on conventional T-type micromixers and those utilizing the SAR or multilamination principle, employing the Villermaux–Dushman test reaction. They evaluated and compared the mixing efficiency of these micromixers, considering the Reynolds number and power dissipation per unit mass of liquid. Furthermore, Viktorov et al. conducted a comparative study of three passive micromixers (teardrop, Y-Y, and H-C) over a wide range of Reynolds numbers [11]. Their study combined numerical simulations and experimental analyses to evaluate the mixing performance at Reynolds numbers between 1 and 100. Furthermore, Bošković et al. [12] examined and compared the residence time characteristics in three distinct passive micromixers. In their study, Ali et al. [13] analyzed the mixing performance of a passive planar micromixer with obstructions at low Reynolds numbers. The micromixer design includes diamond-shaped obstructions in the microchannel, allowing flow breakup and recombination.

Furthermore, a novel passive planar micromixer that incorporates dislocated subchannels was developed by Jian Li et al. [14], exploiting the concept of planar asymmetric separation and recombination for improved mixing efficiency. A comparable investigation was executed by Gidde et al. in 2019 [15], wherein they provided computational and experimental assessments contrasting the mixing behavior of micromixers engineered with diverse forms of impediments. A novel micromixer was introduced, specifically the modified two-layer crossover (TLCM) configuration. This shape was recently employed by Naas et al. [16] to attain superior mixing performance with nanofluid concentrations for a non-Newtonian power-law fluid. Viktorov et al. [17] introduced a novel SAR micromixer termed the H-C mixer, which integrates the designs of the H mixer [18] and the Chain mixer [19].

Many works have been carried out on Newtonian fluids mixing in different microfluidic devices. Studies on non-Newtonian fluids remain limited. Furthermore, most existing research has used active micromixers [20], which have the practical drawbacks mentioned above. The integration of nanofluids is regarded as an effective approach to enhance thermal mixing efficiency and diminish entropy formation in micromixers, hence offering a feasible alternative to traditional thermal systems [21].

Naas et al. [22] examined the properties of steady laminar flow of non-Newtonian nanofluids within a chaotic microdevice, namely TLCCM. A technique was validated for both Newtonian and non-Newtonian flow within a continuously heated geometry under investigation. Moreover, numerous studies [22,23,24,25,26] have emphasized the significant influence of nanoparticle concentration on entropy formation in non-Newtonian fluid flows, suggesting an ideal volume fraction to reduce irreversibility. Ferrofluids have recently gained prominence in the investigation of magnetic micromixers. Cao et al. [27] created a ferrofluid-based microfluidic magnetic micromixer utilizing a hybrid magnetic field generated by micromagnets alongside an external uniform AC magnetic field. Chen et al. [28] constructed a novel micromixer utilizing three-dimensional laminar flow. This structure facilitates the creation of several overlapping fluid layers during constant laminar flow, leading to enhanced mixing efficiency.

Magnetohydrodynamic (MHD) mixing has gained attention as a compelling approach that effectively merges the enhanced mixing performance typical of active systems with the operational simplicity and cost-effectiveness inherent to passive micromixers. By applying an electric and a magnetic field to an electrolyte solution, a Lorentz force is generated that can induce secondary flows for stirring and mixing.

In their work, Nouri et al. [6] studied both numerically and experimentally the mixing of deionized water and Fe_3_O_4_ ferrofluid in a Y-shaped micromixer using a permanent magnet. They conclude that applying a magnetic field considerably improves the mixing efficiency of the micromixer and reduces the mixing length. Tsai et al. [20] experimentally examined mixing of the water and Fe_3_O_4_ ferrofluid in a Y-shaped micromixer. They state that mixing efficiencies can reach over 90%.

The use of non-Newtonian fluids in MHD systems represents a largely unexplored area of research with considerable potential for innovation. Non-Newtonian fluids exhibit complex rheological behaviours that can be exploited to improve mixing and reduce energy consumption.

This research aims to enhance the comprehension of nanofluid-based micromixers and assess their viability as alternatives to conventional thermal systems, thereby facilitating further investigations in this domain. The chaotic flow production and thermal mixing efficacy of the proposed micromixer were evaluated utilizing various nanoparticle concentrations and fluid behavior index values. An assessment of the energy expenditure associated with mixing and fluid index homogenization will be performed to attain substantial energy efficiency and reduce entropy formation. This research seeks to enhance the comprehension of micromixers utilizing nanofluids and assess their feasibility as substitutes for traditional thermal systems, consequently promoting additional investigation in this domain.

The study of non-Newtonian nanofluids in MHD micromixers represents a major advance over conventional studies limited to Newtonian fluids. Combined with magnetohydrodynamics (MHD), non-Newtonian nanofluids introduce a new active control mechanism that optimises chaotic advection, thermal mixing efficiency and entropy generation in micromixers. We analyzed the chaotic flow generation and thermal mixing performance of the proposed micromixer by varying the nanoparticle concentrations and fluid behavior index values under the effect of MHD. In addition, we will evaluate the energy costs of mixing and fluid index homogenization to get enhanced energy efficiency and diminished entropy production.

## 2. Models and Validation

The geometric configuration of a micromixer significantly influences its mixing performance. Cross-junction micromixers typically employ a network of channels intersecting at specific angles, enabling fluid interactions at junctions. Design must target optimal channel dimensions (width, height, length) tailored to the properties of the mixed fluids [29]. A common approach utilizes staggered herringbone structures or chaotic advection principles to enhance mixing.

This study proposes a modified two-layer cross-channel micromixer (MLCC) as illustrated in Figure 1. Naas et al. [16] initially implemented it to enhance mixing performance for power-law non-Newtonian fluids influenced by nanofluid concentrations [21]. Figure 1 depicts the geometry of the TLCM. The micromixer consists of two helical channels, with a periodic chamber created by the configuration of the lower and top channels. Subsequent mixing units are located in multiple recreated grooves. The specified dimensions of the geometry are: *d* = groove diameter (0.8 mm), *I* = distance between inlets (0.8 mm), *D* = chamber diameter (0.2 mm), *L** = geometry length (4.5 mm) and *d_hyd_* = hydraulic diameter (0.22 mm).

### 2.1. Theoretical Framework

#### 2.1.1. Governing Equations

The subsequent equations denote the controlling equations [16] and were resolved numerically with CFD software (ANSYS Fluent 16):(1)$$div\;\vec{V}=0$$
where $\vec{V}$ is the velocity vector.(2)$$\overset{⃑}{V}\cdot \overset{̿}{\nabla }\overset{⃑}{V}=-\frac{1}{\rho_{nf}}\vec{\nabla }P+div\tau$$
where *τ* (Pa) and *P* denote shear stress and pressure, respectively.(3)$$\rho_{nf}c_{nf}\overset{⃑}{V}\cdot \vec{\nabla }T=\lambda_{nf}\Delta T$$
where $T,\;\lambda_{nf}$ and $\rho_{nf}$ are the temperature, thermal conductivity and density of the examined nanofluid.

The relationship between shear rate *γ* (s^−1^) and shear stress *τ* (Pa) can be described by a straightforward power-law equation:(4)$$\tau =m{\dot{\gamma }}^{n}$$
where *n* and *m* represent the fluid behaviour index and consistency index.

The viscosity equation is expressed as:(5)$$\mu_{nf}=k{\dot{\gamma }}^{n-1}$$

The applied boundary conditions include a uniform velocity profile at the inlet; a no-slip condition on all walls; and atmospheric pressure at the exit. For species transport, the mass fraction is set to 1 at inlet 1 and 0 at inlet 2. In thermal mixing simulations, the hot and cold inlets are defined at 330 K and 303 K, respectively, while all walls are considered thermally insulated.

All walls are assumed adiabatic. This simplification isolates the mixing-driven thermal homogenization from external heat loss or gain, allowing for a focused analysis of the internal convective and chaotic mixing performance. While this may not fully represent all real microfluidic devices where conjugate heat transfer occurs, it provides a controlled basis for comparing the fundamental efficiency of the proposed design.

By solving conservation equations characterizing the sources of charge and diffusion for each constituent species, ANSYS Fluent can model the mixing and transport of fluid types. The mass fractions of all types ‘*i*’ add up to one, of course. In order for ANSYS Fluent to determine the local mass fraction Ci of each species, it solves the charge and propagation equation for that species when solving conservation equations for fluid types are chosen.

In its most basic form, the conservation equation looks like this:(6)$$\nabla \;\cdot \;(\rho VC_{i})=-\nabla \;\cdot \;J_{i}$$

$J_{i}$ denotes the mass diffusion of fluid species. This flow results from two mechanisms: the first involves concentration gradients, and the second pertains to pressure gradients that can induce diffuse motion within the mixture.

#### 2.1.2. Fluid Physical Properties

In this study, the non-Newtonian nanofluids used in this simulation, the properties of the fluid in both cases are explained in the following Table 1.

#### 2.1.3. Mass Transfer Characteristics

The efficiency of the mixing process is directly affected by the flow rate via the micromixer. When the Reynolds number is small, less than 1, the main mixing process is diffusion and laminar flow takes center stage. In order to maintain laminar regimes and guarantee enough residence time for optimal mixing, designers must accurately control flow rates [30]. Kozicki et al. [31] suggested a parametric approach to generalize the Reynolds number for flows with complicated cross-sections in the context of power-law non-Newtonian laminar flow. A new Reynolds number, flow, was introduced by them:(7)$${Re}_{g}=\frac{\rho U_{i}^{2-n}D_{h}^{n}}{\left[8^{n-1}{\left(b^{*}+\frac{a^{*}}{n}\right)}^{n}k\right]}$$
where *a** and *b** are set at 0.21 and 0.67, respectively, ρ denotes the fluid density in (kg m^−3^), *n* stands for the power index, *k* stands for the power consistency index, and *U_i_* stands for the input velocity in (m/s).

#### 2.1.4. Mixing Index


(8)
$$MI=1-\frac{\sigma }{\sigma_{0}}$$


The average deviation of the mass fraction specified in [19] is as follows:(9)$$\sigma^{2}=\frac{1}{N}\sum_{i=1}^{N}{\left(C_{i}-\bar{C}\right)}^{2}$$

The input section *σ*_0_ represents the standard deviation (SD), $C_{i}$ is the mass fraction at the inspection point *i*, and $\bar{C}$ is the ideal mixture mass fraction. *N* is the total number of sample points in the cross-sectional segment. The formula below may be used to get the greatest standard deviation for the given data range:(10)$$\sigma_{0}^{2}=\bar{C}\left(1-\bar{C}\right)$$

A balance between input energy efficiency (the amount of energy needed to push the fluid down the channel) and the cost of mixing must be found throughout the estimating phase [29]:(11)$$MEC=\frac{\Delta P\times Q}{MI}$$

*MEC* is here defined as the hydraulic power ΔP×Q required to sustain the liquid motion; the additional electrical power that may be needed to generate the magnetic field is not included, because its magnitude is hardware-specific and lies outside the scope of the fluid-dynamic benchmark.

#### 2.1.5. Kinetic Characteristics

The vortex stretching and compression coefficients, *α*, were determined to define this behavior in the flow. It is delineated by the following phrase [16]:(12)$$\alpha =\frac{\vec{\Omega }\cdot \overset{̿}{D}\cdot \vec{\Omega }}{\Omega^{2}}$$

$\overset{̿}{D}$ represents the strain tensor, whereas $\vec{\Omega }$ denotes the vorticity vector. When *α* > 0, the expansion of the vortex supersedes its contraction [31]. The variations in the average vorticity rate (Ω_mean_) inside the micromixer correspond to a Reynolds number range of 0.5 to 25. This parameter is delineated by the following equations [32]:(13)$$\Omega =\frac{1}{2}{\left[{\left(\frac{\partial w}{\partial y}-\frac{\partial v}{\partial z}\right)}^{2}+{\left(\frac{\partial u}{\partial z}-\frac{\partial w}{\partial x}\right)}^{2}+{\left(\frac{\partial v}{\partial x}-\frac{\partial u}{\partial y}\right)}^{2}\right]}^{\frac{1}{2}}$$(14)$$\Omega_{mean}=\frac{1}{℧}\int \Omega d℧$$
where ℧ denotes the total fluid volume inside the channel. Enhanced flow rates and energy consumption are associated with an elevated mass mixing index.

#### 2.1.6. Heat Transfer Properties

Where the heat flux of the wall is expressed by q″ (w/m^2^), the average temperature of the wall is defined by *T_w_* (K) and the average volume temperature is called *T_b_* (K). The thermal mixing index (*TMI*) quantifies thermal mixing, computed for hot and cold fluids as follows.(15)$$TMI=1-\frac{\sqrt{\frac{1}{n}\sum_{i=1}^{n}\;{\left(T_{i}-\bar{T}\right)}^{2}}}{\sigma_{0}}$$

$\bar{T}$ represents the average temperature at the designated plane, whereas *T_i_* denotes the individual average temperature. They are delineated as follows [33] and must be assessed to achieve a balance between the expense of thermal mixing and its efficacy regarding input power (the inertia necessary to propel the fluid):(16)$$TMEC=\frac{\Delta P\times Q}{TMI}$$

#### 2.1.7. Entropy Generation Principles

By using the flow field temperatures and velocity distribution, we can ascertain the irreversibility of local entropy production resulting from heat transfer ($s_{T}^{\prime \prime \prime }$) and the irreversibility of fluid friction ($s_{p}^{\prime \prime \prime }$) over the three dimensions of the flow [34]:(17)$$s_{T}^{\prime \prime \prime }=\frac{\lambda }{T^{2}}\left[{\left(\frac{\partial T}{\partial x}\right)}^{2}+{\left(\frac{\partial T}{\partial y}\right)}^{2}+{\left(\frac{\partial T}{\partial z}\right)}^{2}\right]$$(18)$$s_{p}^{\prime \prime \prime }=\frac{\mu }{T}\left[2\left({\left(\frac{\partial u}{\partial x}\right)}^{2}+{\left(\frac{\partial v}{\partial y}\right)}^{2}+{\left(\frac{\partial w}{\partial z}\right)}^{2}\right)+{\left(\frac{\partial u}{\partial y}+\frac{\partial v}{\partial x}\right)}^{2}+{\left(\frac{\partial u}{\partial z}+\frac{\partial w}{\partial x}\right)}^{2}+{\left(\frac{\partial v}{\partial z}+\frac{\partial w}{\partial y}\right)}^{2}\right]$$

#### 2.1.8. Governing Equations of Magnetohydrodynamics

The governing equations for an electrically conductive fluid in a magnetic field include the Navier–Stokes equation with the Lorentz force, conservation of mass, conservation of charge, and Ohm’s law [4]:(19)$$\rho V\cdot \nabla V=-\nabla p+\mu \nabla^{2}V+J*B$$(20)∇·V=0(21)∇·J=0(22)J=σ(−∇∅+V∗B)

In this context, V, ρ, p, μ, σ, J, and ∅ represent the velocity vector, fluid density, pressure, dynamic viscosity of the fluid, electrical conductivity of the fluid, current density vector, and electric potential, respectively. In a one-dimensional shear flow of Newtonian fluids, the shear stress is represented as *σ* = μ·du/dy, where the constant μ is referred to as the dynamic viscosity of the fluid.

### 2.2. Numerical Approach

Using ANSYS Fluent 16 © CFD software based on the finite volume method (FVM) [34], all equations governing this work were solved in a laminar flow regime. The SIMPLEC method was selected for the connection of pressure and velocity. The mass and momentum equations were derived with a second-order upwind method. The computations were checked and simulated to converge to residual values of 10^−7^ RMS (root mean square). Non-Newtonian fluids exhibiting power-law behavior were used as working fluids across various concentrations of Al_2_O_3_ nanoparticles. A quantitative grid test was conducted to evaluate the sensitivity of the numerical findings by altering the overall cell count. Four mesh grids, including 100,000 to 800,000 nodes, were analyzed using an unstructured mesh with homogeneous tetrahedral cells, with mesh widths between 0.001 and 0.00011 mm; refer to Figure 2 and Table 2.

The results of numerical simulations are strongly affected by the mesh generation method, which needs a mesh independence test to select the optimal mesh size. In this study, mesh independence tests were performed on four different sizes, with the number of cells varying from 400,000 to 700,000. It can be seen that the pressure loss for meshes with 600,000 and 700,000 cells is very similar, with the maximum error between the two curves being less than 0.048%. Consequently, a grid of 600,000 cells corresponding to a mesh size of 11 μm was adopted for the numerical simulations. Table 2 illustrates the pressure loss of fluids to demonstrate the independence of the mesh.

### 2.3. Validation of Simulation Models

The numerical results’ correctness was verified using numerical simulations of a chaotic micro-mixer with obstacles [35] at a constant Reynolds number, as seen in Figure 3a. The discrepancy in inaccuracy between the findings of Li et al. [36] and our simulations was determined to be markedly below 1%.

## 3. Results and Discussion

The focus is primarily on examining the hydrodynamic and kinetic characteristics of the proposed design. To assess mixing efficiency with MHD effects, the study involves analyzing and visualizing mass transfer profiles, velocity vectors, velocity distributions, and stress points at different levels and locations.

### 3.1. Hydrodynamic Measures for Improving Mixing Performance

In dynamical systems theory, the complexity of flow structures arises at the time when the phase space contains an unstable element. In contrast to elliptic points, these unstable components are known as hyperbolic points. It is evident from this pattern that chaotic advection has begun to form in the flow.

For various nano concentrations, φ = 1, 3 and 5% and Reynolds numbers Re = 5–25, Figure 4 displays the secondary flows in the middle planes. At low Reynolds numbers (creeping flows), the velocity contours still show two big vortices forming in the micromixer, as well as the junction of secondary flows in the plane coming from the top and bottom of the channel. Two fluid jets colliding is the best way to describe the incoming flows that meet. Even at low concentrations, fluid jets significantly improve mixing very quickly.

The cross-sectional vortices get stronger and more clearly defined with increasing Reynolds numbers [37]. The tangential velocity profile (Figure 4) and the radial velocity profile (Figure 5) show the streamline crossings and the uphill and downhill flows, respectively. In addition, in every instance that was considered, the greatest velocity point was found close to the wall. So, the fluids that are sitting on the walls eventually make their way to the middle of the cross-section. The axial velocity profiles reveal that the flow slows down in the duct’s core and speeds up close to the walls on either side. This points to the generation of secondary flows, the significance of which is conditional on the values of the Reynolds number and the behavior of the nanoparticles.

The suggested shape displays the secondary flow vectors of fluid particles in designated areas (chamber inlets and outputs; see Table 3). A recycling zone has been formed by new forms as a result of the chaotic micromixer’s unique geometric feature—the abrupt shift in direction. The result is a distorted main channel and the observation of vectors crossing over each other, leading to a uniform transverse fluid flow; subsidiary flows were also substantial.

Figure 5 shows the effect of Reynolds numbers on the intensity of fluid vorticity in five distinct nanofluid examples. The flow intensity for all micromixers clearly increases with increasing Reynolds number, leading to increased kinetic energy and severe chaotic advection. Consequently, secondary flow and vorticity emerge swiftly as Reynolds numbers increase. Turbulence inside the micromixer becomes more chaotic as *φ* drops, even for a given Reynolds number.

From −1 to +1, helicity may be found. There is total anarchy in the fluid flow at the two extremes, −1 and +1. For various Reynolds numbers and nanofluid concentrations, the dimensionless helicity values of the micromixer are shown in Figure 5b.

### 3.2. Mass Transfer Procedures

Compared to the shear-thinning fluid with *φ* = 1%, the two fluids in Figure 6 show a typically increasing mixing index for all Reynolds numbers, indicating that they become more homogenized and mix much better. At very low Reynolds numbers, the micromixer experiences a diffusion phase that results in much higher mixing index values.

According to Figure 6 and sporadic mini-mixer photos, the liquid gradually separates into several thinner layers as the flow progresses. This results in a much larger contact area between the liquids, which in turn improves the mixing performance. This mini-mixer is efficient because it achieves homogenous mixing at the outlet for all values of χ, independent of the value of the generalized Reynolds number.

A visualization of the vectors and streamlines of the mass fraction and center vortex areas is provided in Table 3 to elucidate the fluid flow structure inside the proposed micromixer. The kinematic behaviour significantly enhances the homogenization of nanofluids.

Figure 7 illustrates that the micromixer has a prominent vortex zone in each corner, which enhances the mixing rate inside the geometry, while exhibiting a little pressure drop at the output section. Moreover, the flow exhibits increased chaos and dynamism attributable to the configuration’s structure and curvature. Furthermore, it is significant that the trajectory line inside the newly chosen micromixer generates a reverse flow pattern, resulting in the formation of robust secondary flows that enhance mass transfer efficiency and guarantee superior homogenization quality.

The influence of highly chaotic advection results in a micromixer fluid concentration mixing level of 93% for Reynolds numbers around 25. Given that nanoparticles alone may homogenize the flow, the Reynolds number is very low (Re < 5), resulting in suboptimal flow behaviors for enhancing mixing, with molecule diffusion prevailing.

As the Reynolds number escalates, homogenization becomes more efficient, and the intensity of mixing advances swiftly. At higher Reynolds numbers, nanoparticles exhibit superior performance, and the mixing intensity escalates more quickly, enabling the concentration to attain the optimal selective mixing state. In comparison to the scenario when *n* = 0.88, the proposed micromixer exhibits a 2.22% increase in mixing intensity when the fluid behavior index decreases to 0.46.

Figure 8 presents an examination of the costs associated with integrating MEC energy across various nanoparticle concentration scenarios for different Reynolds numbers. The anticipated cost of mixing energy is articulated as a function of input power (mW). The flow velocity directly influences the augmentation of flow rate, resulting in an escalation in mixing energy costs across all situations as the Reynolds number rises. Flow velocity influences thermal and hydrodynamic conditions. This arises from the observation that the pressure drop varies at a greater order of magnitude than the mixing index when the Reynolds number escalates. At low Reynolds numbers, namely Re < 10, the energy expenditure for mixing is often reduced.

The cost of mixing energy is significantly influenced by the power law index, with values decreasing as the fluid index or concentration rises. This pertains to the rheological properties of fluids, since an elevation in the power index results in a reduction in apparent viscosity, hence facilitating the agitation of the fluid. Consequently, the energy expenditure for mixing diminishes while the mixing index markedly escalates.

### 3.3. Heat Transfer Procedures

The thermal mixing of fluids in both cold and hot non-Newtonian nanofluids is evaluated across a spectrum of nanofluid concentrations, specifically ranging from ϕ = 0.5 to 5%. In reference to the subsequent figures, this process is achieved by introducing the fluid heated to 330 K through one inlet, while simultaneously injecting the cold fluid at 300 K through the other.

This section examines the energy cost of mining and the thermal mixing capabilities of the current micromixer. Its results are compared to those of more powerful micromixers.

As the Reynolds number increases, the quality of homogenization of the thermal mixture improves for a given value of the fluid behavior index or concentration of the nanofluid. As the quantity escalates, the dynamics of the movement experience significant enhancement, the kinematics of the fluid nanoparticles exhibit considerable variation, and the intensity of the mixture is amplified.

The progression of the standard deviation across various fluid concentrations and Reynolds numbers is presented in Table 4, elucidating the influence of secondary flow generated by vortices on heat transport. The data indicates that uniformity is a significant factor in the vortex regions of the flow. The findings indicate that the mixer serves as a tool for assessing thermal performance across various scenarios. The fluids exhibit a state of complete mixing and demonstrate a tendency to achieve homogeneity as they traverse the geometry within the diffusion regime characterized by Re = 0.5, attributable to the chaotic nature of fluid flow dynamics. It is clear that when *φ* = 5%, the influence on the mixing process is significantly pronounced, resulting in an elevated degree of homogeneity.

Figure 9 illustrates the TMI, a metric assessed in the outlet fluid flow across various Reynolds numbers, considering the impact of nanoparticle concentrations. The definition can be articulated as the quotient of the temperature variance and the mean temperature. The enhancement of thermal mixing capabilities is evident across all Reynolds number scenarios; the peak thermal mixing index is attained at elevated concentrations of nanofluids, with the index nearing 100%, indicating complete mixing.

Figure 10 and Figure 11 illustrate the temperature distribution contours within cross sections and meridian planes of the examined micro-mixers, varying across different behavior indices (*φ* = 0.5–5%) and for two Reynolds numbers (Re_g_ = 5, 15, and 25). Thermal mixing is influenced by the fluid behavior index and the Reynolds number. Figure 11 illustrates that for *φ* = 0.5, the temperature distribution within the meridian plane and across the cross-sections of the micromixers indicates a relatively low mixing performance, particularly at Re = 5, which aligns with the chaotic advection regime. In contrast, at Re = 15, a notable enhancement is evident in these two geometries, with mixing primarily attributed to molecular diffusion. Conversely, the micromixer with a diameter of 5 exhibits superior mixing performance across both flow regimes. In the case of fluids exhibiting a Reynolds number of 20, the performance of mixing is notably elevated, and there is a modest enhancement in mixing quality as the Reynolds number remains at 20.

The thermal efficacy of nanofluids is influenced by several variables. The behavior of nanofluids may create a temperature differential inside the flow, hence enhancing the thermal energy of the nanofluid.

The streamlines merge at the junctions, refer to Figure 11. Fluid from the upper-layer channel flows into the lower-layer channel, integrating with its flow, and vice versa. The appearance of the two differently colored streamlines after traversing the first node of the junction plainly demonstrates a stretching and folding of the fluid interface. The micromixer’s shape induces streamline entanglements at junctions and vortices along the vertical channel sidewalls, facilitating chaotic advection and enhancing mixing efficiency. In the other three geometries, the two fluids interact and coalesce in the primary channel, then proceed in parallel till reaching the channel outlet. A little deviation of the streamlines was seen in the geometry.

Figure 12 illustrates the development of mixing energy costs for various nano-concentrations. In terms of Reynolds numbers, it is evident that the present micromixer exhibits the most efficient mixing energy expenditure when compared to the micromixer previously analyzed by Embarek et al. [29]. Additionally, to quantitatively assess the mixing index, pressure drops, and mixing energy cost, Figure 12 presents several recent studies (2017, 2019, 2021) for these parameters at a constant Reynolds number (Re = 30). The suggested micromixer exhibits the minimal pressure drops, the greatest mixing index value of 99.99%, and the most efficient mixing energy cost.

### 3.4. Entropy Generation Analysis

This section investigates entropy generation and thermal mixing efficiency for a non-Newtonian nanofluid inside an innovative micromixer. Driven by the need for accurate response regulation and sustainable resource management, the research investigates nanofluids as a viable approach to improve thermal efficiency and reduce entropy. The methodology assesses chaotic flow production and mixing efficacy across different nanoparticle concentrations and fluid behavior indicators. The primary goal is to evaluate the energy expenditure associated with mixing and fluid homogenization to enhance energy efficiency and minimize entropy production, hence ascertaining the feasibility of nanofluid-based micromixers as substitutes for traditional thermal systems.

#### 3.4.1. Synergy

The field synergy principle posits that convective heat transmission depends on the scalar product of the velocity vector and the temperature gradient, therefore relating to the orientation of these two vectors. It may thus serve as a valuable instrument to characterize and explain the localized augmentation of heat transport mechanisms. Recent researches have concentrated on a localized examination of the Field Synergy Principle (FSP) by including the impact of the synergy modulus into the investigation.

Nevertheless, little research has investigated the impact of the rotation protocol of solid mixer walls on convective exchanges in constrained spaces. This research intends to use this idea as a metric for assessing heat exchange efficiency in the micromixer and to investigate the influence of vortices generated by the movement of the maize walls on altering synergy angles. This section aims to compute the synergy angle, which is directly associated with the efficiency of heat transmission in the micromixer. The fluctuations in thermal performance of this design are assessed and analyzed based on changes in the arrangement of local velocity fields and temperature gradients, using the local field synergy concept. Refer to Figure 13.

#### 3.4.2. Entropy Generation and Thermal Efficiency of the Studied Micromixers

The local friction entropy production is contingent upon the comparatively modest micromixer geometry of laminar flow. Increased Reynolds values lead to more entropy generation owing to the amplification of inertial forces. Moreover, as seen in Figure 14, the entropy generated by fluid friction is marginally reduced for *φ* = 5% compared to other conditions, attributable to the corresponding increased pressure drop. The impact of varying nanofluid amounts is, however, minimal.

Figure 14 illustrates the spatial distributions of $s_{P}^{\prime \prime \prime }$ for various nanofluid concentration levels at Re = 10.

The maximum entropy production resulting from the irreversibility of $s_{P}^{\prime \prime \prime }$ in the geometries is seen in areas immediately near the wall across all instances of nanofluid concentration. The spatial extent of the layer exhibiting substantial values of $s_{T}^{\prime \prime \prime }$ seems to expand with the rise in the Reynolds number, as seen in Figure 14.

For all cases of non-Newtonian nanofluids, the entropy generation rate increases as the Reynolds number increases for the micromixer.

Figure 15 presents a comparison of $s_{T}^{\prime \prime \prime }$ with varying values of *φ* and Re. Concerning the micromixer, it is apparent that in all instances of non-Newtonian fluids, the irreversibility of thermal entropy production is reduced, and its size reduces with an increase in flow behavior (*φ*).

Heat transfer entropy formation escalates with an increase in Reynolds number (Re). These findings confirm that reducing Re and augmenting *n* may effectively enhance thermal performance in chaotic flow. Minimal nanofluid concentrations result in maximal entropy generation. The findings confirm that the proposed chaotic micromixer may markedly enhance heat mixing efficiency regarding irreversible heat transfer.

For a certain Reynolds number Re = 10, Figure 16 shows how various nanofluid concentration values *φ* = 1, 3, and 5 affect the local distributions of entropy production owing to heat transfer $s_{T}^{\prime \prime \prime }$. In the other locations, $s_{T}^{\prime \prime \prime }$ is almost negligible, but in the corner of the micromixer, there is a distribution of considerable entropy formation owing to heat transfer. Entropy development due to heat transmission is almost nil in other locations and concentrated in a narrow layer next to the wall for all geometries of nanofluids. When the concentration of nanofluids is small, the Reynolds number law generates the most entropy. Nevertheless, in every instance, the formation of entropy owing to fluid friction is much more than that due to heat transfer, suggesting that, under the present flow circumstances, fluid friction irreversibility dominates entropy generation. Hence, improved work performance would result from a fluid system with a low *φ* and a high Re.

Figure 16 and Figure 17 illustrate the relationship between the Reynolds number and entropy generation resulting from heat transfer in the micromixer. The findings are detailed for the scenario where *φ* equals 3 and Re ranges from 0.5 to 25. The analysis reveals that in the context of geometry, the entropy generation resulting from heat transfer irreversibility in all cases of nano-non-Newtonian fluids at low Reynolds numbers is evident, with its magnitude escalating as the flow regime intensifies. With an increase in the Reynolds number, there is a notable decrease in entropy generation associated with heat transfer, further highlighting the significant impact of the Reynolds number on heat transfer performance. The results consistently demonstrate that chaotic flow significantly enhances heat transfer performance by elevating the Reynolds number and reducing non-Newtonian nanofluid concentrations. Refer to Figure 16 and Figure 17 for detailed insights.

### 3.5. Magnetohydrodynamics Behaviours

The present research employs a specifically chosen structured grid system that has been derived from a series of grid independence evaluations. The iteration process for the coupled governing equations incorporates under-relaxation. A second-order upwind scheme is utilized for the discretization of the convective terms, whereas a central difference scheme is employed for the diffusion terms. The approach employed for the discretized equations involves an accelerated multi-grid lower–upper incomplete factorization technique as outlined by Xuejiao et al. [38,40]. This comparative study investigates the effects of MHD on mixing in microfluidic systems. By analysing various magnetic field configurations and intensities, this study highlights how MHD can enhance or hinder mixing processes compared to traditional passive mixing techniques. The results demonstrate that MHD can lead to improved mixing performance through induced vortices and modified flow patterns, offering valuable insights for optimizing microfluidic designs in applications, see Figure 18.

Figure 19 shows the relationships between the mixing rate, the Reynolds number, and the effects of MHD. Conditions of laminar flow are typical in a low regime, where viscous forces are the dominant factor. By applying Lorentz forces to the conducting fluid, MHD may induce secondary flows (such as vortices) that improve mixing.

The thermal or mass mixing rate can improve with increasing magnetic field intensity, as MHD effects become more pronounced. As Re increases, MHD effects can still contribute to mixing but may become less predictable.

At very high Re, the mixing rate may stagnate or even decrease if the magnetic field disrupts the turbulent structures that facilitate mixing. See Figure 19. In general, the mixing strength increases with increasing Re (up to a certain point) and with increasing magnetic field intensity due to enhanced flow structures.

There is an optimal range of Re and magnetic field intensity where mixing efficiency is maximized. The interaction between inertial and viscous forces, as well as MHD effects, creates complex flow dynamics that can be exploited to enhance mixing in microfluidic applications, as shown in Figure 20 and Figure 21.

## 4. Conclusions

The purpose of this effort is to pave the way for future research by increasing our knowledge of nanofluid-based micromixers and determining whether or not they can replace traditional thermal systems. Through the manipulation of nanoparticle concentrations and the fluid behavior index in response to magnetohydrodynamics, this research delves into the creation of chaotic flow and the thermal mixing efficiency of the suggested micromixer.

Comparative analysis of the effects of magnetohydrodynamics on mixing in microfluidic systems is presented in this article. In order to evaluate the degree to which MHD influences mixing in comparison to more traditional passive approaches, it investigates a variety of magnetic field configurations and intensities. Through the generation of vortices and the modification of flow behavior, the results demonstrate that MHD has the potential to improve mixing efficiency. This provides valuable information for the improvement of microfluidic system designs in practically applicable applications.

The optimal performance of the proposed chaotic MHD micromixer is achieved through the combination of key parameters: a *φ* of 4 to 5%, a Re between 10 and 25, and an applied magnetic field corresponding to a Ha of 200 to 300. This configuration maximizes mixing efficiency while maintaining controlled energy consumption.

## Figures and Tables

**Figure 1 micromachines-17-00066-f001:**
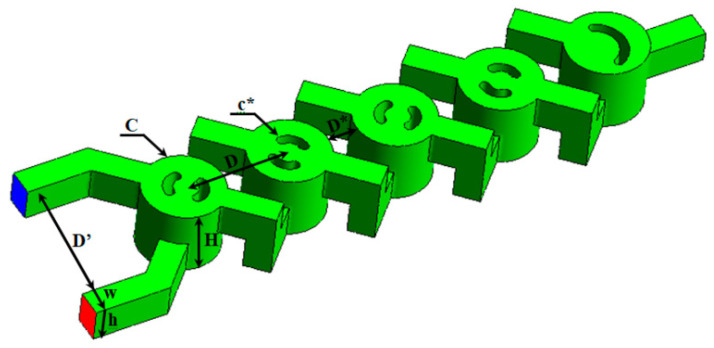
Design depiction of MLCC. H: total height, w: main channel width, h: channel height, C: chamber curvature, D: chamber diameter, D’: inlet center-to-center distance, D*: connecting channel length and c*: groove diameter.

**Figure 2 micromachines-17-00066-f002:**
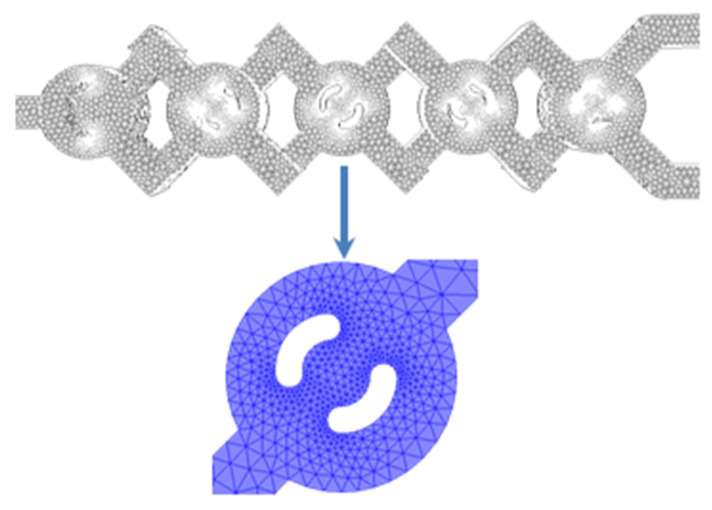
Structure of the generated mesh.

**Figure 3 micromachines-17-00066-f003:**
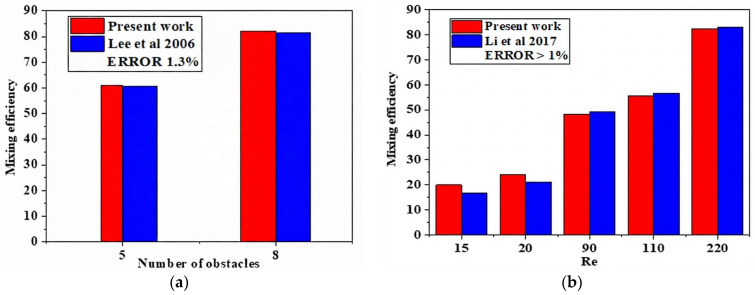
Comparison of mixture efficiency with: (**a**) Lee et al. [35], and (**b**) Li et al. [36]. Furthermore, quantitative numerical validation was conducted utilizing data from Li et al. [36], revealing the heat transfer mixing rate for non-Newtonian cases in relation to varying Reynolds numbers. Figure 3b presents results from a comparison that demonstrated strong agreement between the findings.

**Figure 4 micromachines-17-00066-f004:**
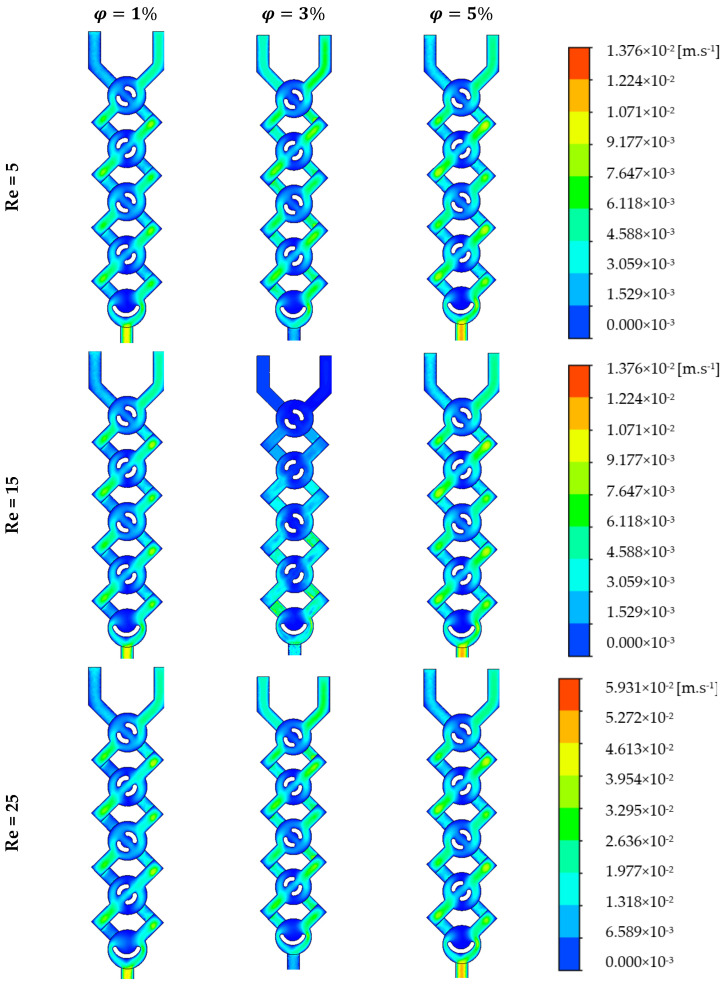
Velocity contours for various *φ* = 1, 3 and 5% Re = 5, 15 and 25.

**Figure 5 micromachines-17-00066-f005:**
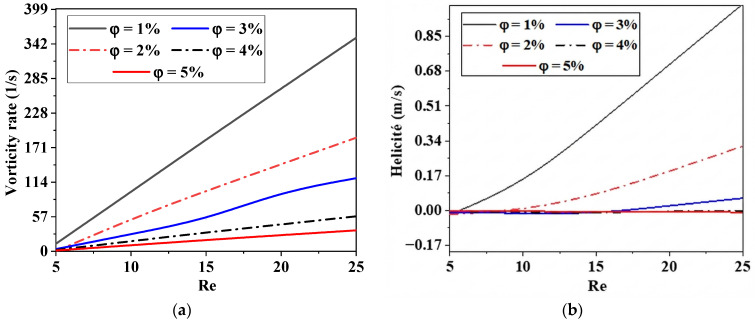
Variations in vorticity intensity. (**a**) Helicity rates (**b**) with Reynolds number (5–25) for different nanofluid concentrations (1, 2, 3, 4 and 5).

**Figure 6 micromachines-17-00066-f006:**
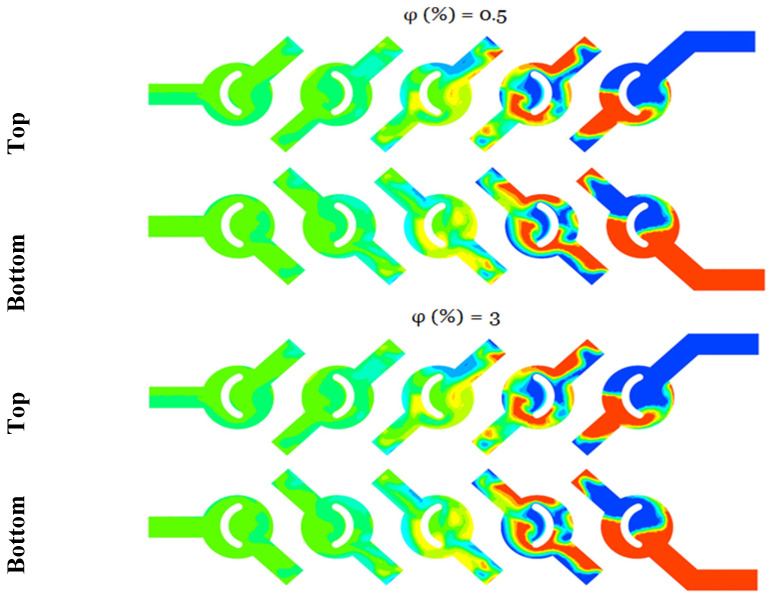
Mean distributions of the planar mass fractions of the micromixer for *φ* = 0.5 and 3, for bottom and top plan.

**Figure 7 micromachines-17-00066-f007:**
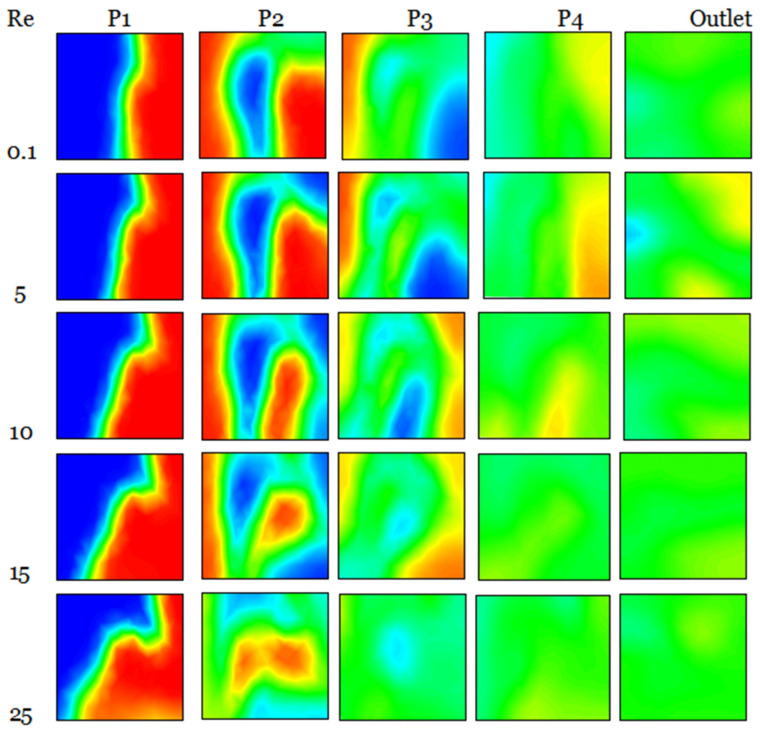
Mass transfer distribution of the micromixer for fluid *φ* = 3%, and Reynolds number 0.1, 5, 10, 15 and 25.

**Figure 8 micromachines-17-00066-f008:**
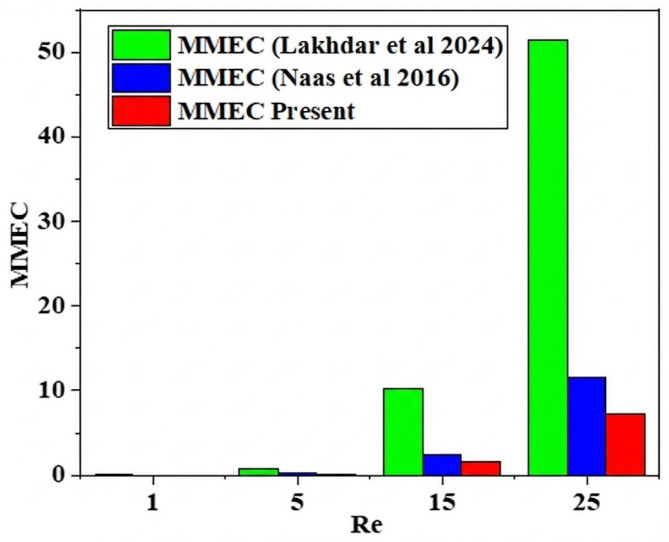
Comparison of the mixing energy cost of micromixers with literatures, for Reynolds number 1, 5, 15 and 25, with that obtained by Lakhdar et al. [38], and Naas et al. [39].

**Figure 9 micromachines-17-00066-f009:**
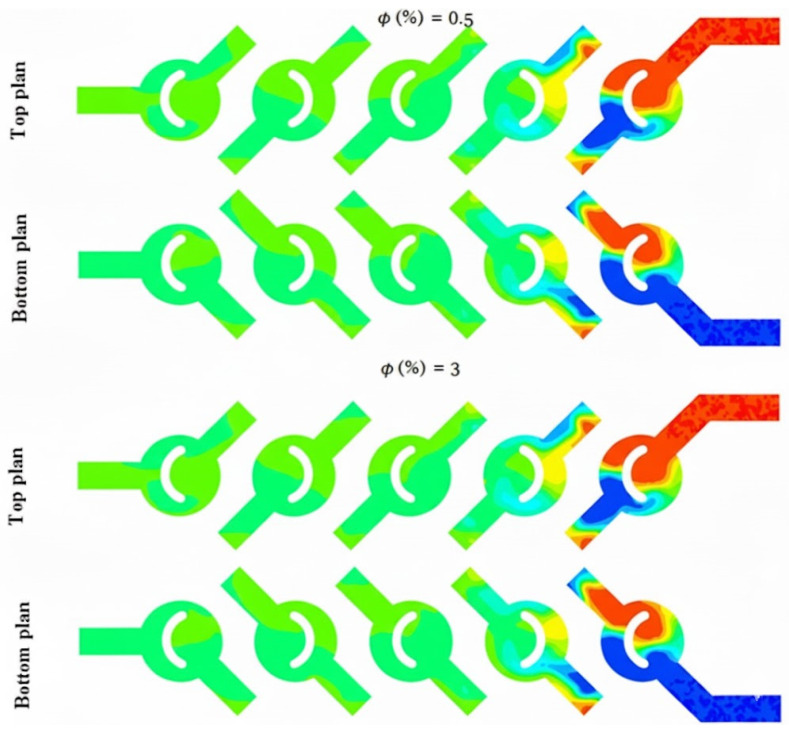
Temperature contours for (*φ* = 0.5 and 3%), for both bottom and top plans.

**Figure 10 micromachines-17-00066-f010:**
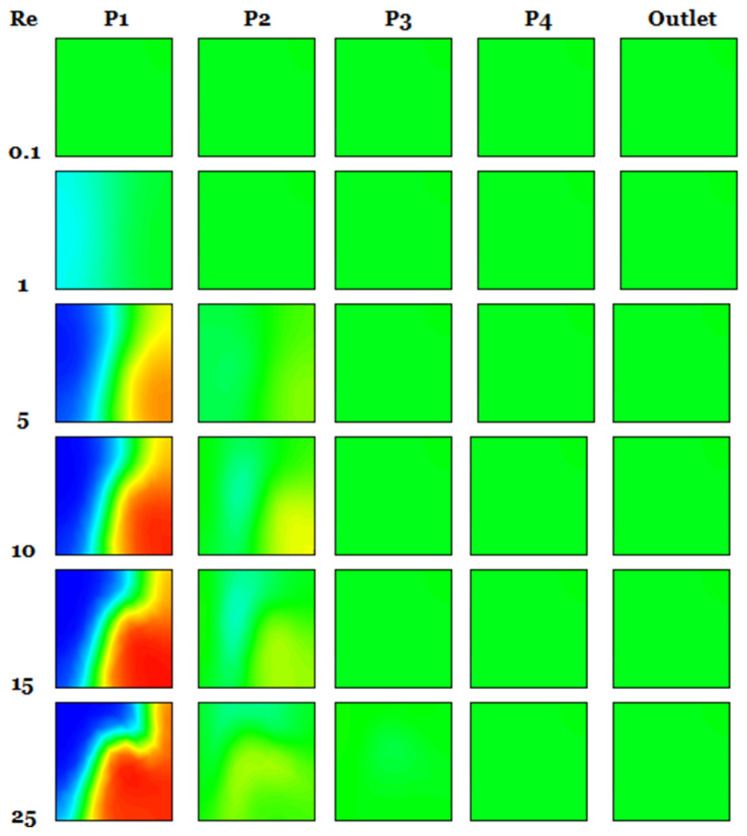
Heat transfer distribution of the fluid micromixer for *φ* = 3% with different Re = 0.1–25.

**Figure 11 micromachines-17-00066-f011:**
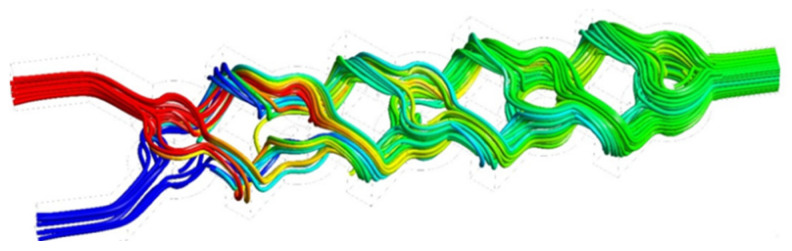
Streamlines coloured by heat transfer for (*φ*% = 5) and Re = 15 of the micromixer.

**Figure 12 micromachines-17-00066-f012:**
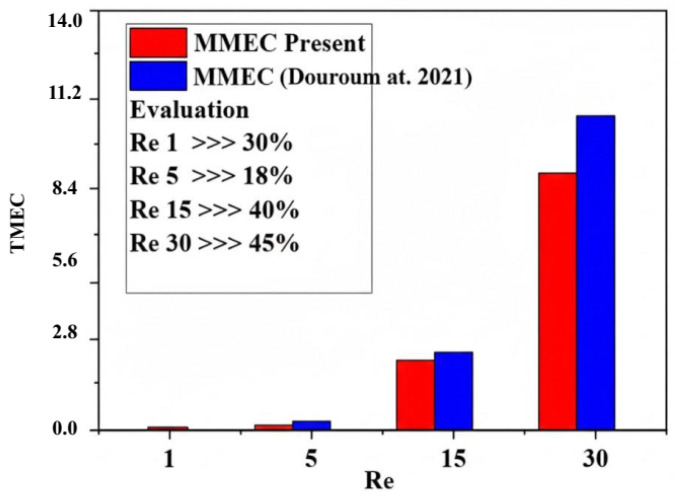
Quantitative comparison of energy mixing cost (TMEC) for various Reynolds numbers (Re = 1, 5, 15 and 30) with that obtained by Douroum et al. [29].

**Figure 13 micromachines-17-00066-f013:**
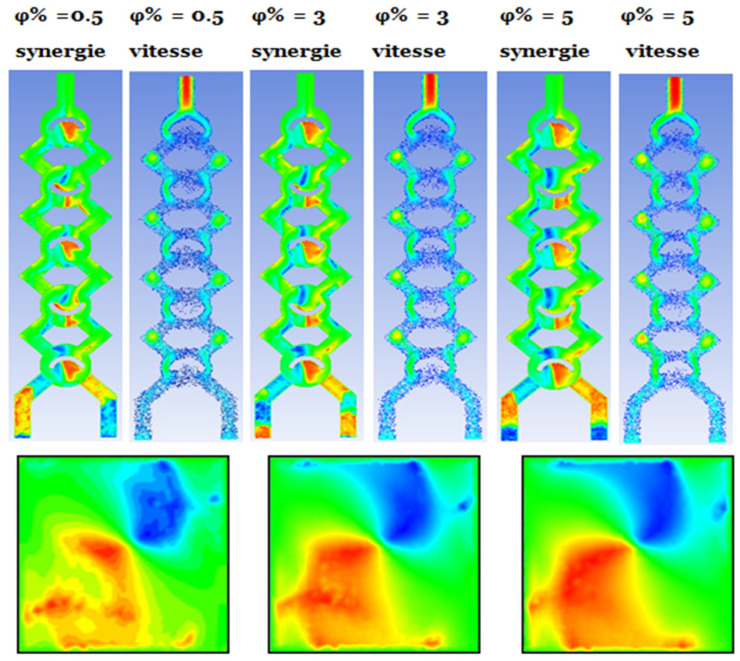
Distribution of the Synergy angle and velocity vector for different concentrations *φ* = 0.5%, 3% and 5%.

**Figure 14 micromachines-17-00066-f014:**
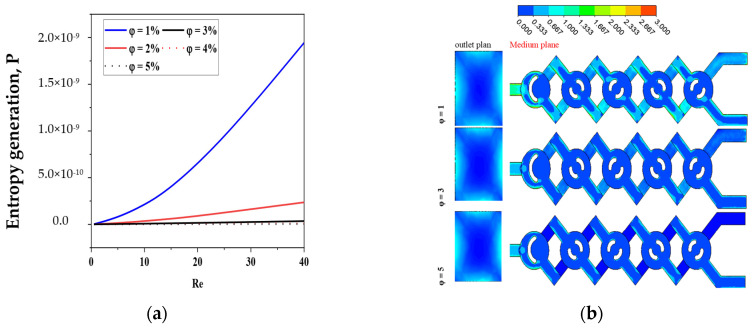
(**a**) Fluid friction entropy generation VS Reynolds numbers under the impact of nano-concentrations *φ* = 1, 2, 3, 4 and 5 with Re = 0.1–40. (**b**) Contour of $s_{P}^{\prime \prime \prime }$ for *φ* = 1, 3 and 5 with Re = 10.

**Figure 15 micromachines-17-00066-f015:**
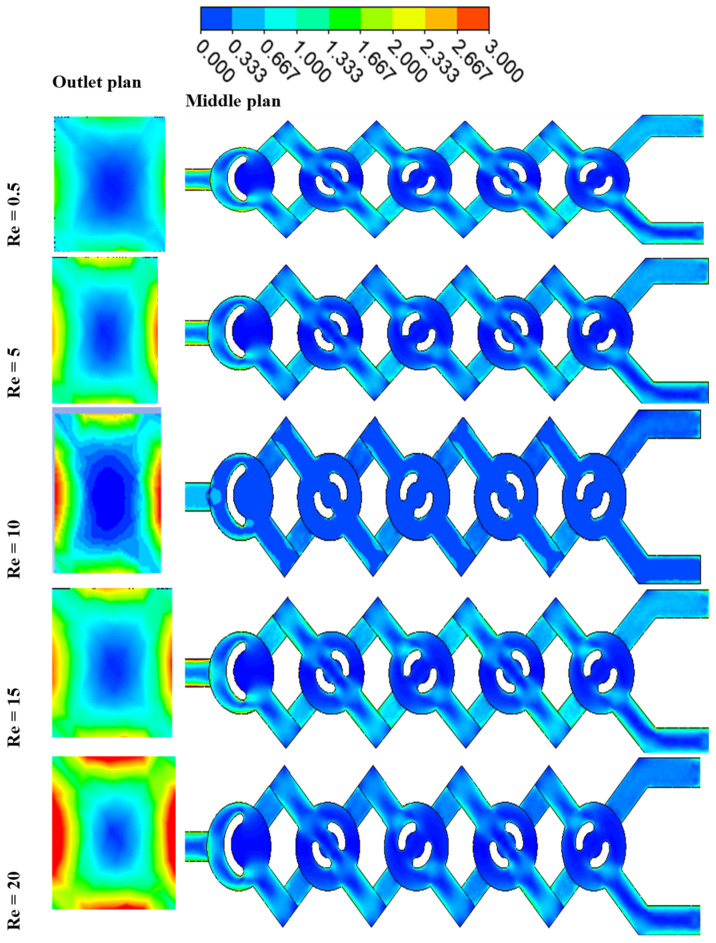
Contour of entropy generation due to friction $\dddot{Sp}$ for Re = 1–20 with φ = 3, for both outlet and middle plans.

**Figure 16 micromachines-17-00066-f016:**
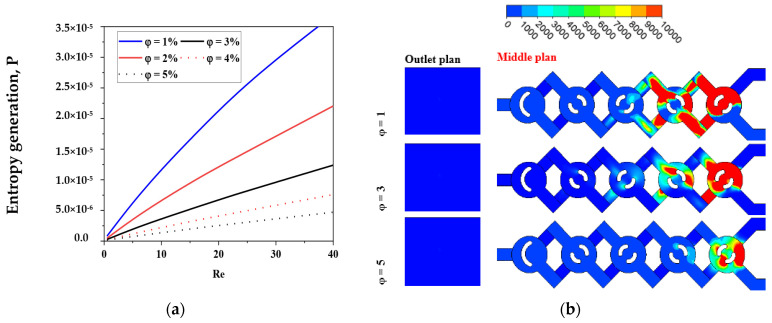
(**a**) Thermal entropy generation VS different Reynolds numbers under the impact of *φ* = 1, 2, 3, 4 and 5 with Re = 0.1–40. (**b**) Entropy generation contour as a function of temperature $S_{T}^{\prime \prime \prime }$ for *φ* = 1, 3 and 5 with Re = 10.

**Figure 17 micromachines-17-00066-f017:**
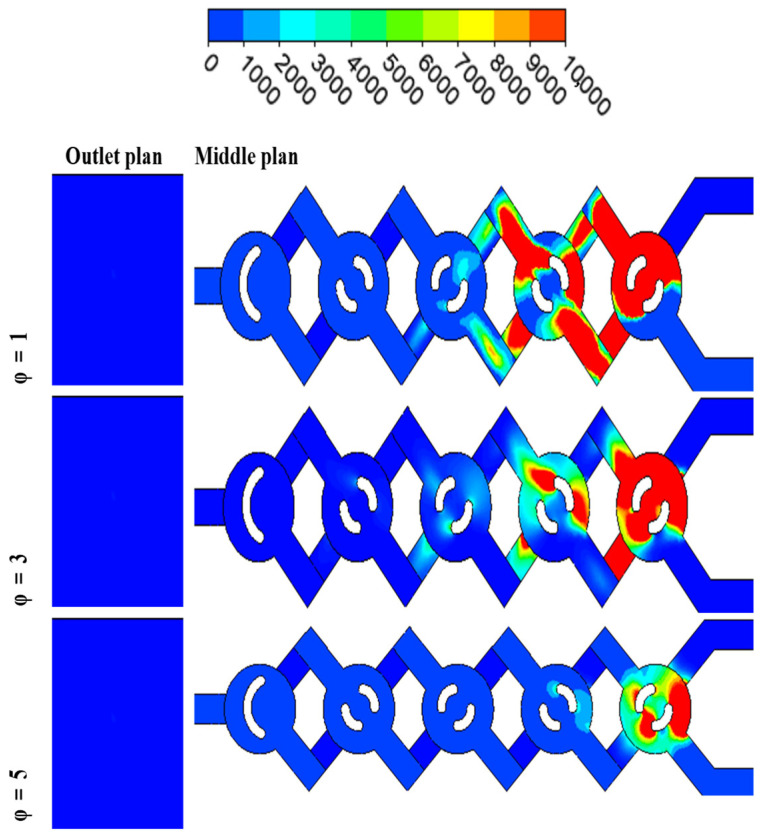
Entropy generation contour as a function of temperature $S_{T}^{\prime \prime \prime }$ for Re = 1–25 with φ = 3%, for both outlet and middle plane.

**Figure 18 micromachines-17-00066-f018:**
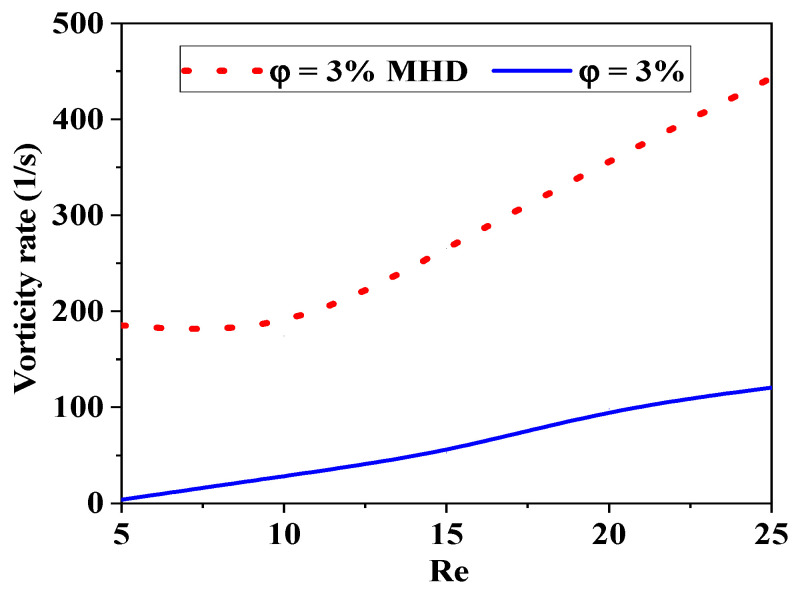
Comparison of kinetic performance efficiency with and without magnetohydrodynamic effects (Ha = 300, *φ* = 3%, Re = 5–25).

**Figure 19 micromachines-17-00066-f019:**
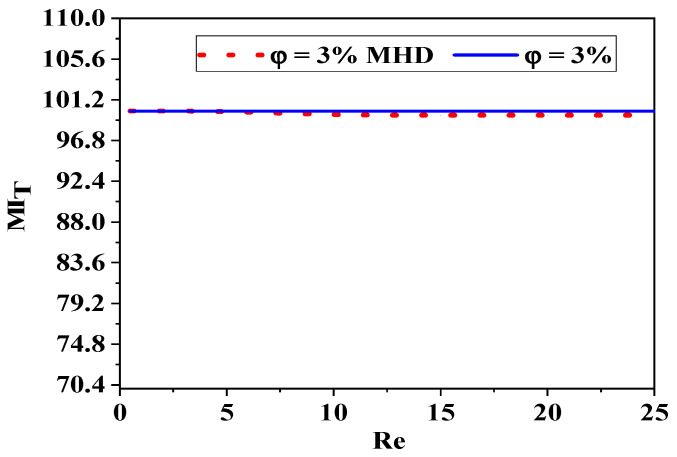
Comparison of mass mixing efficiency with and without magnetohydrodynamic effects (Ha = 200, *φ* = 3%, Re = 0.1–25).

**Figure 20 micromachines-17-00066-f020:**
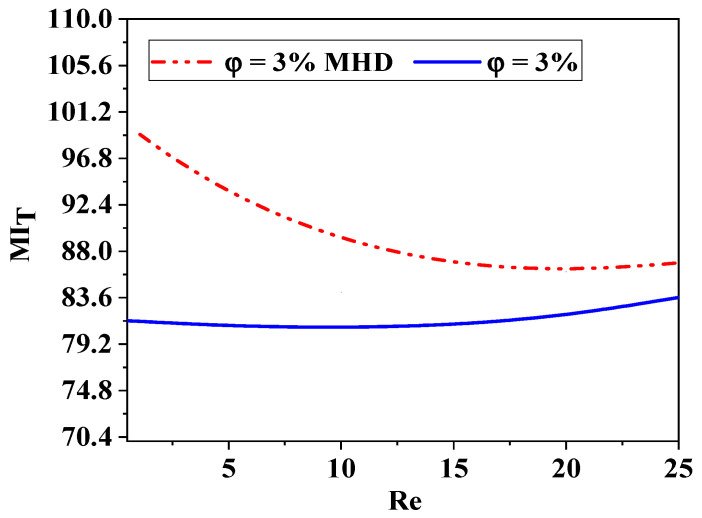
Comparison of thermal mixing efficiency with and without magnetohydrodynamic effects (Ha = 200, *φ* = 3%, Re = 5–25).

**Figure 21 micromachines-17-00066-f021:**
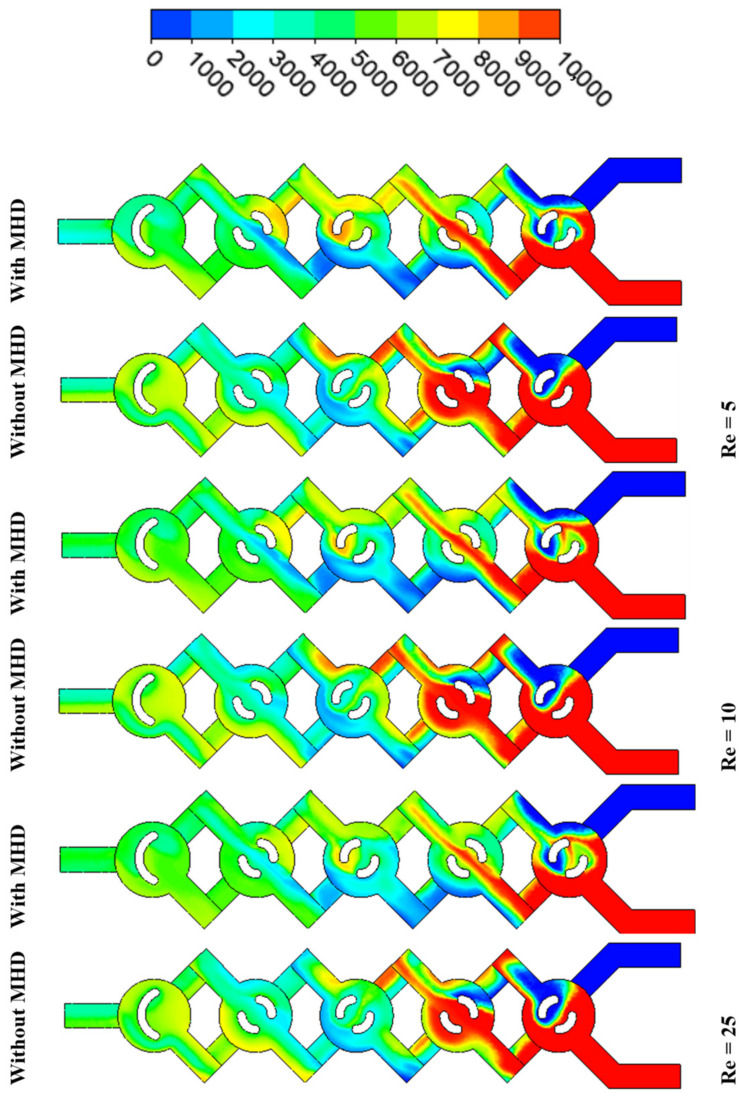
Mass transfer contour for Ha = 0 and Ha = 200 with *φ* = 3%, Re = 5–25).

**Table 1 micromachines-17-00066-t001:** Rheological Parameters (m, *n*) according to *φ* [29].

*φ* %	m (Ns^n^m^−2^)	*n* (-)	$\rho \;(kg\;m^{-3})$	Cp (J kg^−1^ K^−1^)	K (W m^−1^ K^−1^)	µ (Pa s)
1.0	0.00230	0.830	1027.9	4148.8	0.6367	0.0010
2.0	0.00347	0.730	1057.6	4114.6	0.6610	0.0010
3.0	0.00535	0.625	1087.4	4080.5	0.6859	0.0011
4.0	0.00750	0.540	1117.1	4046.3	0.7116	0.0011
5.0	0.01020	0.460	1146.8	4012.1	0.7379	0.0011
1.0	0.00230	0.830	1027.9	4148.8	0.6367	0.0010

where *φ* is volume fraction of solid Al_2_O_3_ nanoparticles in the base fluid, *n* and m represent the flow behaviour index and consistency factor, respectively. A fluid is considered Newtonian when *n* = 1. For non-Newtonian fluids, a value in the range 0 < *n* < 1 indicates pseudo-plastic (shear-thinning) behavior, while *n* > 1 signifies dilatant (shear-thickening) behavior.

**Table 2 micromachines-17-00066-t002:** Sensitivity of the mesh regarding pressure drop.

Mesh Nodes	Pressure Drop (Pa)
400,000	206
500,000	209.5
600,000	211.3
700,000	211.4
Error %	0.047

**Table 3 micromachines-17-00066-t003:** Evolution of the standard deviation of mass for different cases of nanofluid concentration.

Re	φ=1%	φ = 2%	φ = 3%	φ = 4%	φ = 5%
0.5	0.1035	0.0967	0.0929	0.0914	0.0903
5	0.1045	0.1002	0.0954	0.0956	0.0956
10	0.0823	0.0974	0.0963	0.0975	0.0970
15	0.0504	0.0837	0.0950	0.0984	0.0981
20	0.0332	0.0638	0.0907	0.0988	0.0988
25	0.0386	0.0405	0.0819	0.0979	0.0990
40	0.0326	0.0388	0.0433	0.0816	0.0967

**Table 4 micromachines-17-00066-t004:** Evolution of the standard deviation of different cases of mixture temperatures.

Re	φ = 1%	φ = 2%	φ = 3%	φ = 4%	φ = 5%
0.5	2.50 × 10^−5^	0	5.39 × 10^−6^	0	0
10	0.0573	0.0751	0.0438	0.0077	0.0005
15	0.0573	0.0317	0.0833	0.0309	0.0041
20	0.0526	0.0176	0.0927	0.0612	0.0135
40	0.1704	0.0331	0.0089	0.0980	0.0839

## Data Availability

The original contributions presented in this study are included in the article. Further inquiries can be directed to the corresponding authors.

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
