# Peer review of "Impact of Magnetohydrodynamics on Thermal Mixing Efficiency and Entropy Generation Analysis Passing Through a Micromixer Using Non-Newtonian Nanofluid"

_micromachines, 2025, doi:10.3390/mi17010066_

Round 1
Reviewer 1 Report
Comments and Suggestions for Authors
In this work, non-Newtonian nanofluid was used to investigate the impact of magnetohydrodynamics on thermal mixing efficiency and entropy generation analysis passing through a micromixer. But I suggest that the paper not be accepted due to following reasons.
Q.1: Avoid using abbreviations and acronyms in title, abstract and headings.
Q.2: Abstract: Abstract should be written due to some essential points such as research purposes, research methods, research contents and research effects, otherwise the innovation and necessity of the manuscript will not be reflected effectively.
Q.3: 1. Introduction (1)The reader can not understand why the non-Newtonian nanofluid was used to solve impact of magnetohydrodynamics on thermal mixing efficiency and entropy generation analysis passing through a micromixer. So, please, re-focus why the non-Newtonian nanofluid is interesting and overall necessary to be studied and which is the novelty that authors are proposing. (2)Avoid lumping references as in [1-3], [7-9], [10–12] and all other. Instead summarize the main contribution of each referenced paper in a separate sentence.
Q.4: “2. Theoretical Framework, Design, and Configuration of the Micromixer” & “3 Numerical approach” & “ 4. Validation of simulation models” (1)“2. Theoretical Framework, Design, and Configuration of the Micromixer” & “3 Numerical approach” & “ 4. Validation of simulation models” should be merged into "2. Models and validation". (2)Reviewer assumes that there is not novelty in the section. Is it right? Please, detail which is the novelty. (3)The majority figures are qualitative, not quantitative; for scientific studies, quantitative analysis and generalization are very important. (4)The overall English expression should be improved.
Q.5: 5. Results and Discussion (1)The capability and meaning of this study should be re-estimated. (2)The majority figures are qualitative, not quantitative; for scientific studies, quantitative analysis and generalization are very important. (3)There has been a great effort with some numerous research papers on field synergy principle. Potential references on field synergy principle should include.
Reviewer 2 Report
Comments and Suggestions for Authors< !--StartFragment -->
This study investigates the steady laminar flow of non-Newtonian nanofluids in a chaotic micromixer. The authors demonstrate that chaotic micromixing, enhanced by Al₂O₃ nanoparticles and magnetohydrodynamic effects, achieves nearly rapid thermal mixing efficiency in non-Newtonian nanofluids at low Reynolds numbers.
In the introduction, the authors present various mixing approaches; however, it would be beneficial to add:
- A discussion of the advantages and disadvantages of existing approaches (currently, the authors only state that Author A developed one method and Author B developed another).
- An explanation of why the proposed approach is needed, both conceptually (e.g., energy efficiency) and from an application standpoint.
The authors appear to be extending their previous work, such as "Lakhdar A, Skander J, Tayeb NT, Mostefa T, Hossain S, Kim SM. Analysis of Entropy Generation for Mass and Thermal Mixing Behaviors in Non-Newtonian Nanofluids of a Crossing Micromixer. Micromachines 2024;15:1392", maintaining the same structure but adding nanoparticles and magnetohydrodynamic effects. I wonder whether, in Equation 11, the energy efficiency formulation includes the contribution from the external magnetic force. Additionally, what is the mixing efficiency achieved solely by adding nanoparticles?
Overall, the authors have extended their previous work. The results appear improved, but the significance of the contribution should be emphasized more clearly.
< !--EndFragment -->
Reviewer 3 Report
Comments and Suggestions for Authors
This paper investigates thermal mixing efficiency and entropy generation in a chaotic micromixer using non-Newtonian nanofluids under the influence of magnetohydrodynamics (MHD). The authors employ CFD simulations with ANSYS Fluent to analyze the effects of Reynolds number, nanoparticle concentration, and fluid behavior index on mixing performance, heat transfer, energy cost, and entropy generation. The topic is relevant and innovative in the field of microfluidics and thermal-fluid engineering, and incorporating MHD effects adds an interesting dimension. However, there are areas that require improvement in clarity and presentation. I commented as follows;
1.Clarity and Structure
Abstract;
Lacks quantitative details. For example, “mixing rate close to 99%” is good, but specify the exact conditions.
Introduction;
The distinction between this work and prior studies on MHD micromixers should be emphasized more clearly.
Figures and Captions;
Figures 4-21 need more descriptive captions, including conditions (Re, φ, Ha).
2. Technical Concerns
Boundary Conditions;
All walls are assumed adiabatic, which may not reflect real microfluidic devices. State this limitation explicitly.
MHD Model Details;
Magnetic field strength (Hartmann number) and orientation are not sufficiently discussed. Are these values experimentally feasible?
Entropy Generation Discussion;
Results are presented, but design guidelines for optimal conditions should be clearer.
Minor comments
Discuss the potential for experimental validation.
Provide a clear design guideline by summarizing the optimal range of Re, φ, and Ha for best performance.
Include a trade-off analysis between energy cost and mixing efficiency, ideally with a graphical representation.
Revisions should be required.
Reviewer 4 Report
Comments and Suggestions for Authors
The paper presents a numerical investigation of steady laminar flow and thermal mixing of non-Newtonian nanofluids in a chaotic micromixer. The parametric study covering Reynolds number, nanoparticle concentration, and power-law index is relevant and well defined. The analysis of hydrodynamic and thermal performance is generally sound and informative. With minor revisions for clarity and presentation, the manuscript is suitable for publication.
- Introduction and Background
- The introduction does not provide sufficient background on magnetohydrodynamics (MHD). This concept should be clearly defined and its abbreviation properly introduced.
- The first paragraph in the introduction and Lines 123-128 also lack appropriate references.
- Formatting and Writing Style
- Lines 172–178: These lines should be merged into a single paragraph.
- Line 149: The reference format is not consistent with the rest of the manuscript.
- Section 2.1 (e.g., Line 164): The word “Where” should be changed to “where,” since it continues the sentence after the equations and should not be capitalized.
- When referring to tables and figures throughout the manuscript, ensure they are consistently capitalized (e.g., Table 1, Figure 2).
- Tables
- Table 1: Clearly define all parameters (e.g., density, specific heat, thermal conductivity, viscosity) and provide their corresponding units. The volume fraction is not defined before the table and should be introduced. Also, remove the parentheses around volume fraction in the table.
- Table 2: Please clarify what the reported error represents and explain it in the manuscript.
- Table 4: If precision is not critical, reduce the number of significant digits after the decimal to four.
- Figures
- Place the labels (a) and (b) at the top of the corresponding figures.
- Figure 1: Improve the color contrast of the labels. For example, D* is currently black and difficult to read.
- Figure 2: The quality of mesh figure is not sufficiently clear and should be improved.
- Figures 15 and 21: Remove the red underline beneath “Re” and “Without”.
Round 2
Reviewer 1 Report
Comments and Suggestions for Authors
The authors had addressed the reviewers' comments properly, I recommend this paper be accepted for publication.
Author Response
Thakyou.
Reviewer 2 Report
Comments and Suggestions for Authors
In the original paper mentioning the mixing energy cost (Journal of Fluids and Structures 65 (2016) 1–20), the author mentioned "In our case, this input power is the needed both for running the channel and for heaving the square cylinder.", and the equations were in the same paper. As such, I believe, taking account into only running the fluidic and exclude the external magnetic force may not be proper in this case. The equation 11 may be used for passive micromixing with only fluidic flow is the input power though such as Nishu, Israt Zahan, and Mst Fateha Samad. "Modeling and simulation of a split and recombination-based passive micromixer with vortex-generating mixing units." Heliyon 9.4 (2023).). Alternatively, authors may rephrase mixing energy cost and explain why the authors excluded external magnetic energy or comment about the magnetic energy etc.
L663 5. Patents, does this mean author filed patents?
Reviewer 3 Report
Comments and Suggestions for Authors
The revisions are satisfied.
Author Response
Thankyou.
Round 3
Reviewer 2 Report
Comments and Suggestions for Authors
The authors have addressed all of my concerns, and the manuscript has been sufficiently improved for publication.